# Association between Low Energy Availability (LEA) and Impaired Sleep Quality in Young Rugby Players

**DOI:** 10.3390/nu16050609

**Published:** 2024-02-23

**Authors:** Oussama Saidi, Maher Souabni, Giovanna C. Del Sordo, Clément Maviel, Paul Peyrel, Freddy Maso, Fabrice Vercruyssen, Pascale Duché

**Affiliations:** 1Laboratory Youth-Physical Activity and Sports-Health (JAP2S), Toulon University, F-83041 Toulon, France; oussama.saidi@univ-tln.fr (O.S.); maher.souabni.phd@gmail.com (M.S.); clement.maviel@univ-tln.fr (C.M.); fabrice.vercruyssen@univ-tln.fr (F.V.); 2Interdisciplinary Laboratory in Neurosciences, Physiology and Psychology—Physical Activity, Health and Learning (LINP2), Paris Nanterre University, F-39200 Nanterre, France; 3Psychology Department, New Mexico State University, 1780 E University Blvd, Las Cruces, NM 88003, USA; giovanna.del@hotmail.fr; 4Department of Kinesiology, Laval University, Quebec City, QC G1V 0A6, Canada; paul.peyrel.1@ulaval.ca; 5Quebec Heart and Lung Institute, Laval University, Quebec City, QC G1V 0A6, Canada; 6Rugby Training Center of the Sportive Association Montferrandaise, F-63100 Clermont-Ferrand, France; fmaso@asm-omnisports.com

**Keywords:** sport nutrition, energy intake, energy expenditure, polysomnography, adolescents, rugby, relative energy deficiency in sport

## Abstract

Low energy availability (LEA) has been associated with several physiological consequences, but its impact on sleep has not been sufficiently investigated, especially in the context of young athletes. This study examined the potential association between energy availability (EA) status and objective sleep quality in 42 male rugby players (mean age: 16.2 ± 0.8 years) during a 7-day follow-up with fixed sleep schedules in the midst of an intensive training phase. Participants’ energy intake was weighed and recorded. Exercise expenditure was estimated using accelerometry. Portable polysomnography devices captured sleep on the last night of the follow-up. Mean EA was 29.3 ± 9.14 kcal·kg FFM^−1^·day^−1^, with 47.6% of athletes presenting LEA, 35.7% Reduced Energy Availability (REA), and 16.7% Optimal Energy Availability (OEA). Lower sleep efficiency (SE) and N3 stage proportion, along with higher wake after sleep onset (WASO), were found in participants with LEA compared to those with OEA (*p* = 0.04, *p* = 0.03 and *p* = 0.005, respectively, with large effect sizes). Segmented regression models of the EA-sleep outcomes (SE, sleep onset latency [SOL]), WASO and N3) relationships displayed two separate linear regions and produced a best fit with a breakpoint between 21–33 kcal·kg FFM^−1^·day^−1^. Below these thresholds, sleep quality declines considerably. It is imperative for athletic administrators, nutritionists, and coaches to conscientiously consider the potential impact of LEA on young athletes’ sleep, especially during periods of heavy training.

## 1. Introduction

Restorative sleep is considered the gold standard strategy for recovery after training and competitions [1]. However, inconsistent sleep schedules, insufficient sleep duration, and poor sleep quality, including prolonged sleep onset latencies (SOL), frequent awakening after sleep onset (WASO), and decreased sleep efficiency (SE), are prevalent problems among team sports athletes, particularly younger ones [2,3,4,5]. This has been associated with an increased risk of injury, burnout, and concussion, especially in those involved in contact sports such as rugby [6,7,8]. For this very reason, the identification of modifiable factors associated with altered sleep patterns in young athletes is of paramount importance [9]. Although studies have focused on sport-related factors such as training (volume, timing), competition, or travel affecting sleep [10,11], few studies to date have attempted to examine how energy status influences athletes’ sleep.

In this context, it is well established that adolescence is a life stage during which dietary requirements for energy, macronutrients, and micronutrients increase due to the demands of growth and development processes [12,13]. Given the high energy expenditure associated with training, it could be even more challenging for young athletes to achieve adequate energy intake. Several studies have reported that athletes competing in team sports, including rugby, often fail to meet recommended dietary guidelines for their sport and activity level [14,15,16,17], especially during training camps, where they undergo high training loads, sometimes exceeding 15 h of training per week [14,18,19]. In this context, research has shown that low energy availability (LEA)—where an individual’s dietary energy intake is insufficient to support the energy expenditure required for health, function and daily living, once the costs of exercise and sporting activities are taken into account [20,21]—negatively impacts an individual’s psychological state [22] and physical [22] and cognitive [23] performance.

The recently published International Olympic Committee Consensus Statement on REDs supports the notion that decreased sleep quality may be an indicator associated with LEA [24]. However, there are a limited number of studies that have examined the relationship between LEA and sleep. More importantly, the reported results are inconsistent. For instance, Pardue et al. [25], in a case report study, evaluated the effects of contest preparation (8 months where caloric intake was gradually decreased), followed by recovery (5 months), on a competitive drug-free male bodybuilder over 13 months. The results showed that during the preparation, cortisol and ghrelin increased while subjective sleep quality decreased. However, minimal changes were reported in sleep as measured via accelerometry. In another study, lightweight rowers described difficulties sleeping (i.e., waking up hungry during the night or early morning and going to sleep early in the evening to avoid the feeling of hunger) during a restricted calorie intake and increased energy expenditure program [26]. Recently, LEA and sleep quality have been investigated in a cohort of male Army Reserve Officer Training Corp students. The results showed no significant correlation between energy availability and sleep quality [27]. However, this study only used a subjective assessment of sleep quality among a limited number of participants (*n* = 13). Thomas et al. [28] explored sleep patterns over a prolonged period of time, including chronic (8 weeks: phases 1 and 2) followed by acute (5 days: phase 3) weight loss practices in a case report concerning a taekwondo athlete. They found a decrease in total sleep time (TST) but an increase in SE during the third phase compared to phases 1 and 2. In addition, daily EA status was not significantly associated with objectively measured sleep outcomes. However, it is important to note that the athlete was in an LEA state throughout the entire study period and that this study did not document sleep before or after the weight loss practices, which precludes comparison to baseline sleep. More importantly, the time available for sleep in this case report may have been influenced by early sport-related schedules, as mentioned by the authors.

To the best of our knowledge, no studies have focused on EA and sleep in young athletes. Moreover, in view of the inconclusive results, it is important to underline that most of the above-mentioned studies used subjective measures of sleep, and no study has reported measuring sleep staging using polysomnography (PSG). More importantly, the association between EA and sleep may have been biased by sport-related factors. For instance, we previously showed that increased time spent on sport-related activities resulted in decreased opportunities for sufficient sleep [29]. In addition, the relationship between LEA and sleep is not yet clear. Therefore, the aim of this study is to investigate the relationship between EA status and sleep quality measured via ambulatory PSG under a fixed time spent in bed among young rugby players. We hypothesize that an LEA state might have a negative impact on sleep parameters.

## 2. Materials and Methods

### 2.1. Participants

Forty-two adolescent rugby players engaged in the under-18 national categories (late- or post-pubertal: Tanner stages 4 or 5) participated in this study. They were in good health and were not taking any medications that might interfere with their sleep outcomes (i.e., antidepressants and benzodiazepines). They were not part of any dietary restriction program and did not consume tobacco, cannabis, or alcohol. All diagnosed sleep disorders were considered exclusion criteria, and all participants were screened for obstructive sleep apnea before the start of the study using the Berlin questionnaire [30].

This study complied with the ethical principles of the Declaration of Helsinki and was approved by the relevant Institutional Ethics Review Board prior to the study launch (IRB-00012476-2021190394). Participation in the study was voluntary; parents and young athletes were informed of the purpose of the study and could withdraw at any time. Information and consent forms were distributed to both parents and young athletes before the study began.

### 2.2. Study Setting and Procedure

At baseline, the participants were assessed for anthropometric and body composition (Tanita, MC-780MA S, Tokyo, Japan) and took part in habituation night sleeping with the portable sleep devices (Sleep Profiler, PSG2, Advanced Brain Monitoring, Carlsbad, CA, USA). Then, they participated in a 7-day follow-up at their training center (Montferrand Sports Association-Rugby Section, Clermont Ferrand, France) as well as their respective boarding school (habitual sleep environment). The follow-up took place during an intensive training phase outside of the competitive season (no competition was held). The weekly training load was 15 h/week and included rugby field training for 8 h/week, wrestling for 4 h/week, and strength and conditioning training for 3 h/week. Physical activity was monitored through the use of accelerometry. The participants were instructed to consume food at will in their habitual catering facilities (school and training center canteens). All foods were weighed by the investigators before and after consumption using an electronic food scale. Caffeinated beverages such as cola, coffee, or tea were not allowed after noon. During the nights, the athletes were given an opportune time for 9 h of sleep from 10:30 p.m. to 07:30 a.m. They were asked not to use any electronic media (smartphone, laptop, etc.) in the evening. During the last night of the follow-up, they were instructed to go to their room at 10:00 p.m. and were equipped with sleep devices. The investigators made sure that the sleep devices were properly placed and that the lights were turned off at 10:30 p.m. (Figure 1).

### 2.3. Measurements

#### 2.3.1. Anthropometric and Body Composition Measurements

All of the anthropometric measurements were conducted in the morning in a fasting state by the same specialist. Barefoot height was measured using a portable stadiometer (Tanita, HR001, Tokyo, Japan). Body mass (BM) and composition were measured using a bioelectrical impedance analyzer (Tanita, MC-780MA S, Tokyo, Japan). Body mass index (BMI) was calculated as BM (kg)/height^2^ (m^2^).

#### 2.3.2. Energy Intake (EI)

Dietary intake over the 7-day follow-up period was analyzed using the gold standard of the weighed and recorded food method. Then, a professional computerized nutrient analysis program (Bilnut 4.0 SCDA Nutrisoft software, Cerelles, France), which was supplemented with details of local foods (the French food composition database [ANSES, 2020]), was used to analyze the data. The analysis permitted the calculation of energy intake (in kcal) from dietary intake-derived macronutrients (carbohydrate, fat, and protein in grams).

#### 2.3.3. Exercise Energy Expenditure (EEE)

During the 7-day period, EEE was estimated using tri-axial accelerometry (GT3X+, Actigraph LLC, Pensacola, FL, USA). Accelerometer placement on the dominant hip was achieved using an elastic band over the anterior spine of the iliac crest and aligned with the anterior axillary line of the dominant hip, according to the manufacturer’s guidelines. The participants were instructed to wear the device continuously each day during the follow-up period, except for periods during contact with water. The data were analyzed in 30 s epochs using the manufacturer’s software (Actilife 6.0, Pensacola, FL, USA). MET intensity thresholds were determined based on previous calibration studies and the cut-points established by Evenson et al. [31], which have already been used in studies concerning young athletes. These were chosen because of their higher accuracy compared to other published cut-points [31,32].

#### 2.3.4. Energy Availability (EA)

EA was calculated as the amount of dietary energy remaining after EEE, normalized to fat-free mass (FFM) [24,33]. The EA equation is presented below, where EI is the energy intake and EEE represents exercise energy expenditure. The participants were allocated into three groups based on their EA levels: optimal EA (OEA: EA ≥ 45 kcal·kg FFM^−1^·day^−1^), low EA (LEA: EA < 30 kcal·kg FFM^−1^·day^−1^), and reduced EA (REA: 30 ≤ EA < 45 kcal·kg FFM^−1^·day^−1^) [21].
EA (kcal·kg FFM^−1^·day^−1^) = [EI (kcal·day^−1^) − EEE (kcal·day^−1^)]·FFM (kg)^−1^

#### 2.3.5. Sleep Assessment

Ambulatory PSG (Sleep Profiler PSG2, Advanced Brain Monitoring, Carlsbad, CA, USA) has been validated as a reliable alternative to laboratory PSG for individuals aged 6 and above [34,35]. This portable system provides access to 13 channels, including electroencephalography (EEG), electrooculography (EOG), and electromyography (EMG) at front-polar sites (AF7-AF8, AF7-Fpz, and AF8-Fpz), enabling the comprehensive characterization of sleep staging and quality. Additionally, it records wireless oximetry, nasal pressure/airflow, chest and abdomen respiratory effort, forehead and finger pulse rate, head movement and position, and quantitative snoring. The recorded data were uploaded to the “Sleep Profiler Portal” by Advanced Brain Monitoring in Carlsbad, CA, USA. Automated algorithms were applied to the signals, enabling auto staging through the use of ratios of power spectral densities and the autodetection of cortical and microarousals, sleep spindles, and ocular activity. Subsequently, a sleep expert, blinded to sample identities, reviewed the recordings to confirm the accuracy of the automatic sleep staging [36].

The device provides measurements of total sleep time (TST), sleep onset latency (SOL), wake up after sleep (WASO), sleep efficiency (SE), awakenings longer than 30 s, awakenings longer than 90 s, arousal index, and sleep architecture (N1, N2, N3, and REM sleep) in accordance with the recommendations of the American Academy of Sleep Medicine (AASM) [37].

### 2.4. Statistical Analysis

Statistical analyses were performed using R Studio (version 4.3.2, RCore Team, 2020). Graphing and visualization were carried out using Prism 9 (GraphPad, San Diego, CA, USA). Descriptive statistics are presented as mean ± standard deviations and inferences drawn at a 0.05 alpha level unless otherwise specified. One-way independent ANOVAs were performed to compare the three energy availability groups (OEA, REA, and LEA) on anthropometric and sleep-related variables. When significance was reached, post hoc pairwise comparisons with Tukey’s correction were computed. The normality and homogeneity of variance assumptions were tested (using Shapiro–Wilk and Levene’s tests, respectively). If the assumptions were violated, a logarithmic transformation was applied to the dependent variable. Effect sizes were obtained using the measure of eta-squared (η^2^) for the results of the ANOVAs (the effects were interpreted as follows: <0.01, trivial; 0.01–0.06, small; 0.06–0.14, moderate; and <0.14, large) and Cohen’s d for pairwise comparisons (the effects were interpreted as follows: <0.2, trivial; 0.2–0.5, small; 0.5–0.8, moderate; and >0.8, large). Piecewise/segmented regression involves fitting different regression models to different segments of the data. Each segment has its own set of regression coefficients. Because it was hypothesized that the relationship between sleep outcomes and EA was not constant between EA groups (different intercepts and slopes), piecewise regression models were fitted to identify potential breakpoints where the relationship between the variables may change. Breakpoint values adjusted R-squared and regression coefficients are reported for each model.

## 3. Results

### 3.1. Participants Characteristics

The characteristics of the athletes are presented in Table 1. The participants were allocated into three groups according to their EA status (i.e., OEA, REA, and LEA). Total mean EI, EEE, and EA were 3551 ± 567 kcal, 1552 ± 196 kcal, and 29.3 ± 9.14 kcal·kg FFM^−1^·day^−1^, respectively. Almost half of the participants (47.6%) presented an LEA status, while only 16.6% had an OEA status. No differences in terms of age, height and FFM were detected between the groups. However, higher BM and BF were found in the LEA compared to the OEA group.

### 3.2. Sleep Outcomes among Participants by Energy Availability Status

The sleep outcomes according to EA status are presented in Figure 2. With a 9 h opportunity for sleep, the players slept an average of 7.95 h sleeping per night. Statistical analysis revealed a significant difference in the proportions of SE, WASO, N2 and N3, with large size effects. Post hoc tests showed higher WASO (*p* = 0.005, d = 1.46, Δ = 35%) in LEA compared to OEA. Moreover, time spent in and percentage of N3 stage (*p* = 0.004, d = 1.47, Δ = 17.7 min, *p* = 0.03, d = 1.16, Δ = 3%; respectively) were significantly lower in LEA compared to OEA.

### 3.3. Relationship between Energy Availability (EA) and Sleep Outcomes

The segmented regression models of the EA–sleep outcome relationships displayed two separate linear regions and produced a best fit using a breakpoint between 21 and 33 kcal·kg FFM^−1^·day^−1^. For TST, the marginal adjusted R^2^ = 0.65 (Figure 3a). The model yielded a breakpoint at EA = 22.78 kcal·kg FFM^−1^·day^−1^, which means that the values that are below or above this breakpoint have a different regression line. If EA ≤ 22.78, the equation is as follows: TST = 336.37 (the intercept) + 6.27 (the estimate for EA) × the value for EA. If EA ≥ 22.78, the equation is as follows: ST = 336.37 + 6.27 × (22.78) + (6.27 + −5.97) × (EA − 22.78). For SE, the marginal adjusted R^2^ = 0.65 (Figure 3b). The model yielded a breakpoint at EA = 22.78 kcal·kg FFM^−1^·day^−1^. If EA ≤ 22.78, the equation is as follows: SE = 62.29 (the intercept) + 1.16 (the estimate for EA) × the value for EA. If EA ≥ 22.78, the equation is as follows: SE = 62.29 + 1.16 × (22.78) + (1.16 + −1.11) × (EA − 22.78). For SOL, the marginal adjusted R^2^ = 0.32 (Figure 3c). The model yielded a breakpoint at EA = 23.51 kcal·kg FFM^−1^·day^−1^. If EA ≤ 23.51, the equation is as follows: SOL = 62.85 (the intercept) + −1.93 (the estimate for EA) × the value for EA. If EA ≥ 23.51, the equation is as follows: SOL = 62.85 + −1.93 × (23.51) + (−1.93 + 2.07) × (EA − 23.51). For WASO, the marginal adjusted R^2^ = 0.58 (Figure 3d). The model yielded a breakpoint at EA = 21.86 kcal·kg FFM^−1^·day^−1^. If EA ≤ 21.86, the equation is as follows: WASO = 149.999 (the intercept) + (−4.85) (the estimate for EA) × the value for EA. If EA ≥ 21.86, the equation is as follows: WASO = 149.999 + −4.85 × (21.86) + (−4.85 + 4.42) × (EA − 21.86). For N3, the sleep marginal adjusted R^2^ = 0.21 (Figure 3e). The model yielded a breakpoint at EA = 33.39 kcal·kg FFM^−1^·day^−1^. If EA ≤ 33.39, the equation is as follows: N3 = 64.53 (the intercept) + 1.21 (the estimate for EA) × the value for EA. If EA ≥ 33.39, the equation is as follows: N3 = 64.53 + 1.21 × (33.39) + (1.21 + −1.01) × (EA − 33.39).

## 4. Discussion

EA, accounting for energy intake relative to the energy expenditure associated with exercise, establishes a crucial framework for both health and the efficacy of sports nutrition strategies. Therefore, during an intensive training period for young athletes, EA needs to be satisfied to maximize adaptation to training [38]. However, according to the present study, nearly half of the participants (47.6%) showed EA levels below 30 kcal·kg FFM^−1^·day^−1^ (mean EA = 22.8 ± 4.16 kcal·kg FFM^−1^·day^−1^) during an intensive training phase (15 h/week). These results are in line with previous cross-sectional studies that have reported that low energy availability (LEA) is a widespread phenomenon in young male athletes. For instance, Koehler et al. [39] explored the EA of 167 athletes (mean age = 16.2 years) engaged in a variety of sports (e.g., ball sports, racquet sports, water sports, etc.). The results showed that the EA of 56% of this population was below 30 kcal·kg FFM^−1^·day^−1^ (mean EA = 21.7 ± 6.4 kcal·kg FFM^−1^·day^−1^). Moreover, a high percentage (42%) of collegiate male cross-country runners were not achieving 30 kcal·kg FFM^−1^·day^−1^, as reported by McCormack et al. [40]. Interestingly, the major result of the current study was that the LEA group (EA ≤ 30 kcal·kg FFM^−1^·day^−1^) showed higher WASO and lower N3 stage compared to the OEA group when given the same opportunity for sleep. These results suggest that EA might be directly related to sleep quality. Accordingly, LEA could have a negative impact on objective sleep outcomes (i.e., WASO, SE and cortical arousal).

There is mounting evidence that LEA is a powerful stressor that triggers marked hormonal, metabolic, and psychological responses in male athletes [24,41,42]. It has been shown that even short periods (3–5 days) of LEA trigger acute endocrine and metabolic dysregulations that, when maintained over long periods of time, are believed to result in adverse health and functional outcomes [43,44]. Decreased leptin [39,45,46] and increased ghrelin [25,46] have been widely reported in LEA states. Both leptin and ghrelin are involved in appetite regulation (fasting/feeding) and in behavioral (reward-related) food intake to ensure energy homeostasis and are thought to play a key role in the regulation of the sleep and wakefulness cycle [47]. Ghrelin, predominantly produced in the stomach, stimulates feeding via the activation of orexin-producing neurons, which are implicated in maintaining wakefulness [48]. In contrast, leptin is an anorexigenic adipose hormone. Therefore, a plausible explanation for the increases in WASO and cortical arousal is that, in LEA states, orexigenic signals could be activated in the brain, promoting wakefulness and inducing fragmented sleep and early awakening [49]. A previous study by Falkenberg et al. [50] observed an inverse association between the length of the last meal to bedtime window and sleep duration in professional rugby players during intensive training, suggesting that early awakening could result from low energy flux and higher hunger sensations. Indeed, the theory that humans are wired to stay alert when hungry, helping them to find food, is endorsed. There is an assumption that during prolonged periods of energy unavailability, neuronal populations (mainly orexin) involved in both sleep–wake and metabolic pathways may favor arousal over sleep for foraging and survival purposes [51,52]. Thereby, sleep disturbances in LEA might be a physiological regulation intended to help restore optimal energy flux since previous studies have demonstrated that sleep curtailment led to increased energy intake [53].

Some studies conducted in controlled laboratory settings suggested that short periods of LEA (3–5 days) could be the underlying cause of dysregulations in glucose metabolism [45,54,55]. Koehler et al. [45] manipulated energy intake and expenditure such that each of the six participants completed two conditions of low energy availability (15 kcal·kg FFM^−1^·day^−1^) and two conditions of normal energy availability (40 kcal·kg FFM^−1^·day^−1^) for 4 days in trained men. The authors reported that LEA lowered insulin (−34% to −38%) and fasting glucose (−8% to −12%). In another study, Kojima et al. [54] reported that 3 consecutive days of endurance training under LEA (15 kcal·kg FFM^−1^·day^−1^) decreased fasting blood glucose and muscle glycogen content in seven well-trained male (mean age = 19.9 ± 1.1 yo) long-distance runners. In this context, the interrelationship between sleep and glucose metabolism has been discussed in a recent review [56], in which the authors highlight (i) a possible alteration to the sleep architecture depending on the timing of carbohydrate ingestion and (ii) the need for strict regulation of peripheral glucose since the brain is particularly vulnerable to disruptions in energy supply. Interestingly, it has been reported that a decrease in glucose levels triggers awakening during nighttime sleep in healthy subjects [57,58]. This could be a potential explanation for the altered sleep quality in the LEA group compared to the OEA group. Our piecewise/segmental regression models revealed the existence of a breakpoint (i.e., 22.781, 22.781, 23.511, and 21.863 for SE, TST, SOL, and WASO, respectively) below which sleep quality deteriorates dramatically. According to our prediction model, when EA decreases from 20 to 15 kcal·kg FFM^−1^·day^−1^, SE decreases by 5.8%, and SOL and WASO are almost doubled (from 24 to 34 min and from 53 to 77 min, respectively), which results in a decrease in TST by more than half an hour (31.4 min) in the context of a sleep opportunity time of 9 h. This effect exceeds the threshold viewed as indicative of a clinically meaningful effect on sleep quality, underscoring the significance of these findings. Interestingly, while the universal cut-off of 30 kcal·kg FFM^−1^·day^−1^ at which females experience RED-S-related symptoms is still debated, according to the latest IOC consensus statement [24], such a cut-off or range as a threshold of LEA leading to RED-S outcomes in males appears to be lower (e.g., ~9 to 25 kcal·kg FFM^−1^·day^−1^). This observation is in line with our results, suggesting a cut-off for male adolescent athletes around 23 kcal·kg FFM^−1^·day^−1^. Unlike studies carried out on women, a very limited number of controlled trials ([45,54,55]) have focused on men using a single LEA value (i.e., 15 kcal·kg FFM^−1^·day^−1^). Thus, further research is needed using different LEA values (e.g., 35, 30, 25, and 20 kcal·kg FFM^−1^·day^−1^…) in order to give an insight into the hormonal mechanisms involved in the regulation of sleep from which to draw solid conclusions.

To the best of our knowledge, this is the first study to examine the association between EA and sleep as measured via ambulatory PSG in adolescent athletes. Nonetheless, it is imperative to acknowledge certain limitations. Firstly, there is the potential underestimation of energy expenditure when using accelerometry, particularly in a sports context such as rugby, which warrants consideration. Subsequent research may benefit from employing the doubly labeled water (DLW) method for a more precise assessment of this parameter. Secondly, there is a need to study the relationship between EA and sleep, especially in athletes who participate in weight category sports and in female athletes who are more susceptible to the presence of LEA. However, the strengths of the current study, such as its ecological setting and robust evaluation of energy intake and sleep, make it more interesting. Lastly, it is essential to acknowledge that certain underlying pathways necessitate further exploration. Future studies should investigate appetite sensation and hormonal markers, such as leptin, ghrelin, orexin, etc., to confirm the mechanisms at play.

## 5. Conclusions

The importance of sleep is undeniable, allowing for better physical [59], cognitive [60], and physiological responses, especially in young athletes. The findings of the present study suggest a threshold of LEA at 23 kcal·kg FFM^−1^·day^−1^ in adolescent athletes, below which sleep quality deteriorates dramatically. The fact that low energy availability (LEA) is associated with sleep disturbances underscores the importance for athletic administrators, nutritionists, and coaches to be aware of its potential impact on sleep quality, particularly during periods of intense training. Given the significant role of sleep in athletic performance and overall well-being, proactive measures should be taken to monitor and mitigate the effects of LEA on athletes’ sleep patterns. Collaboration among stakeholders in athlete support networks is essential to implement tailored interventions and strategies to optimize both energy availability and sleep quality, ultimately improving athletes’ overall health and performance outcomes.

## Figures and Tables

**Figure 1 nutrients-16-00609-f001:**
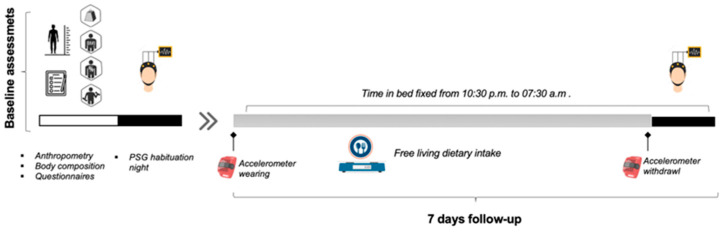
Study overview.

**Figure 2 nutrients-16-00609-f002:**
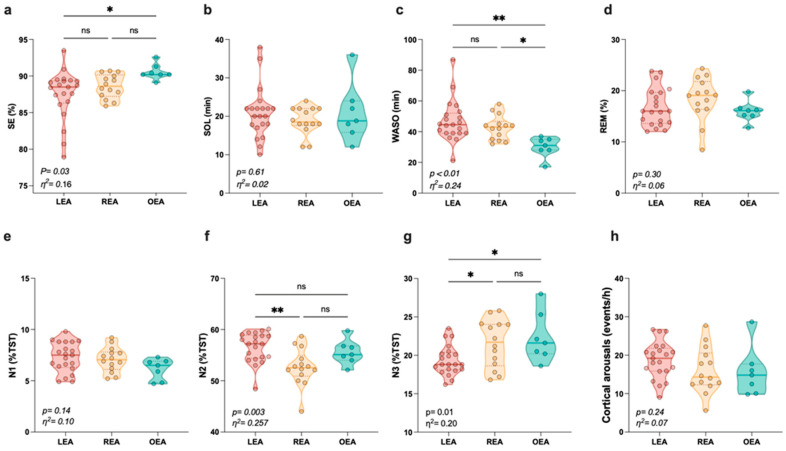
Sleep outcomes among participants in terms of energy availability status. (**a**) Sleep efficiency, (**b**) Sleep onset latency, (**c**) Wake after sleep onset, (**d**) REM, (**e**) N1, (**f**) N2, (**g**) N3, (**h**) Cortical arousals. Data are shown as scatter violin plots (truncated) with median and interquartile ranges (the 25th and 75th percentiles). *p* values were determined via one-way ANOVAs to assess the differences between the groups (LEA, REA, and OEA). When significance was reached, Tukey’s post hoc tests were used to assess the differences between groups, *p* value: ns: non-significant, * *p* < 0.05, ** *p* < 0.01. LEA: low energy availability; REA: reduced energy availability; OEA: optimal energy availability; SE: sleep efficiency; SOL: sleep onset latency; WASO: wake after sleep onset; REM: rapid eye movement sleep.

**Figure 3 nutrients-16-00609-f003:**
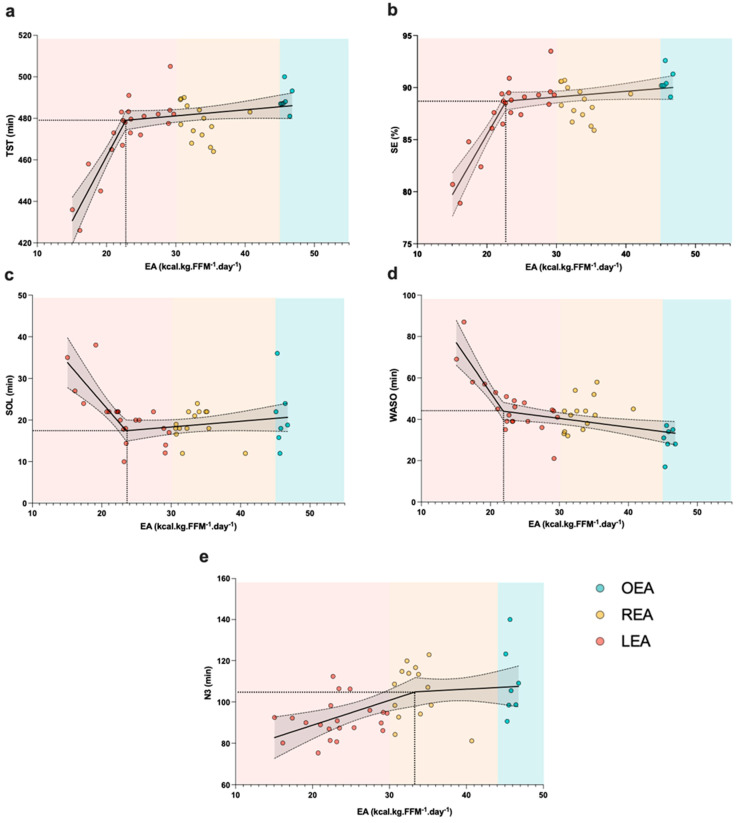
Relationship between energy availability and sleep outcomes using segmented regression models with random coefficients. The full black line represents the mean values (*y*-axis) for each corresponding energy availability (EA) value (*x*-axis), and the broken black lines represent the 95% confidence intervals (CI); dotted lines indicate the breakpoint established by the model. The model detects a breakpoint at (**a**) EA = 22.78 kcal·kg FFM^−1^·day^−1^ for total sleep time (TST), (**b**) EA = 22.78 kcal·kg FFM^−1^·day^−1^ for sleep efficiency (SE), (**c**) EA = 23.51 kcal·kg FFM^−1^·day^−1^ for sleep onset latency (SOL), (**d**) EA = 21.86 kcal·kg FFM^−1^·day^−1^ for wake after sleep onset (WASO), (**e**) EA = 33.38 kcal·kg FFM^−1^·day^−1^ for N3 sleep with different regression line below and above the breakpoint. LEA: low energy availability; REA: reduced energy availability; OEA: optimal energy availability.

**Table 1 nutrients-16-00609-t001:** Athletes’ characteristics (*n* = 42 Males).

	Total	LEA (*n* = 20) EA < 30 kcal·kg FFM^−1^·day^−1^	REA (*n* = 15) 30 ≤ EA < 45 kcal·kg FFM^−1^·day^−1^	OEA (*n* = 7) EA ≥ 45 kcal·kg FFM^−1^·day^−1^	*p*	η^2^
Age (year)	16.2 (0.88)	16 (0.74)	16.4 (0.97)	16.2 (1.12)	0.389	0.04
Height (cm)	181 (6.76)	180 (6.72)	182 (6.77)	183 (7.48)	0.643	0.02
BM (kg)	82.2 (11.3)	86.6 (11.9) ^a^	79.3 (9.19)	73 (4.84) ^a^	0.007	0.20
BMI (kg·m^−2^)	25.1 (3.40)	26.7 (3.36) ^ab^	24.1 (2.76) ^b^	21.9 (0.76) ^a^	0.001	0.28
BF (%)	15.8 (5.36)	18.8 (5.13) ^ab^	13.1 (3.46) ^b^	10.9 (1.56) ^a^	<0.001	0.38
FFM (kg)	68.8 (7.54)	69.7 (8.50)	68.8 (6.70)	65.8 (5.29)	0.493	0.03
EI (kcal·day^−1^)	3551 (567)	3164 (314) ^ab^	3811 (448) ^bc^	4373 (149) ^ac^	<0.001	0.64
EI (kcal·kg^−1^·day^−1^)	44.3 (10.5)	37.3 (6.55) ^ab^	58.5 (6.71) ^bc^	60.1 (4.14) ^ac^	<0.001	0.65
EEE (kcal·day^−1^)	1552 (196)	1583 (185) ^a^	1589 (150) ^b^	1360 (233) ^ab^	0.016	0.17
EA (kcal·kg FFM^−1^·day^−1^)	29.3 (9.14)	22.8 (4.16) ^ab^	32.3 (4.88) ^bc^	45.8 (0.61) ^ac^	<0.001	0.80

LEA: low energy availability; REA: reduced energy availability; OEA: optimal energy availability; BF = body fat; BMI = body mass index; FFM = fat-free mass; EI: energy intake; EEE: exercise energy expenditure; EA: energy availability. When significance was reached, Tukey’s post hoc tests were computed to assess the differences between groups: values sharing the same letter are significantly different, with *p* < 0.05.

## Data Availability

The data presented in this study are available on request from the corresponding author.

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
