# Peer review of "Association between Low Energy Availability (LEA) and Impaired Sleep Quality in Young Rugby Players"

_nutrients, 2024, doi:10.3390/nu16050609_

Round 1

Reviewer 1 Report

Comments and Suggestions for Authors

This is a relational study investigating the association between energy status (energy deficiency and energy sufficient) and sleep quality. The results indicate energy deficiency for these 16 years old athletes disrupts there sleep and caused low quality sleep by several parameters. The study was interesting; however, the authors need to add the full spell of the abbreviations in the figure legends to let the authors clearly see the study from the figures. 

Reviewer 2 Report

Comments and Suggestions for Authors

Abstract:

·         The abstract clearly states the study's purpose, methods, and key findings. However, it could initially benefit from a more explicit statement of the research question or hypothesis.

·         The abstract provides specific and technical details about the methods (e.g., accelerometry and polysomnography), which is excellent for scientific accuracy. Yet, it might be slightly technical for a broader audience. Consider balancing technical details with layman's terms, especially for multidisciplinary readers.

·         The results clearly highlight the significant findings about energy availability and sleep quality. Including the specific statistical outcomes (like p-values or effect sizes) could enhance the reader's understanding of the significance of the results.

·         The study addresses a gap in the research, as stated, but the abstract could emphasize the importance of this research in the broader context of sports science and adolescent health.

1. Introduction

·         The flow from discussing the general importance of sleep in athletes to the specific focus on LEA and its potential impacts is logical and well-structured. However, the transition between these sections could be smoother to enhance readability.

·         The introduction does a good job of narrowing the focus from a broad topic (sleep in athletes) to the specific issue at hand (the impact of LEA on sleep quality). Yet, it could benefit from a more explicit statement of the hypothesis or specific research questions the study aims to address.

·         The section cites various studies to support its statements, showing a comprehensive literature review. However, ensuring the most recent and relevant studies are included could strengthen the introduction's authority.

·         While the introduction sets up the study's background well, it could clearly and concisely articulate its primary objectives and potential implications. This would give readers a more immediate understanding of the study's significance.

2. Materials and Methods

·         The selection criteria and characteristics of the participants are well-defined, enhancing the study's reliability. However, the authors might consider discussing the generalizability of the findings to other populations or sports.

·         The study setting, procedure, and timeline are clearly described, providing a good understanding of how the research was conducted. The authors might benefit from discussing potential limitations or biases introduced by the study's design or setting.

·         The manuscript details the instruments and methods for measuring body composition, energy intake, energy expenditure, and sleep quality. The use of validated tools and methods strengthens the study's validity. However, it would be beneficial if the authors discussed the limitations or potential sources of error associated with these methods.

3. Results

·         The results section is well-structured, beginning with the presentation of participant characteristics, followed by sleep outcomes among participants based on energy availability status, and concluding with an exploration of the relationship between energy availability (EA) and sleep outcomes. This logical flow aids in reader comprehension.

·         Using tables and figures to present the data is appropriate and aids in visualizing the results. However, it is recommended that the authors ensure all figures and tables are referenced in the text at relevant points to guide the reader's understanding.

·         The statistical analysis seems robust, with appropriate use of ANOVAs, post-hoc tests, and regression models. Including effect sizes and confidence intervals adds to the strength of the analysis.

Specific Comments:

1. Participants Characteristics (3.1):

·         The presentation of participant characteristics is comprehensive, and the allocation of participants into groups based on their EA status is well-executed.

·         The authors present an interesting finding that a significant portion of participants presented with low energy availability (LEA), which sets a good stage for further discussion on the implications of this status.

2. Sleep Outcomes Among Participants by Energy Availability Status (3.2):

·         The relationship between EA status and sleep outcomes is presented. The findings that LEA status is associated with higher WASO and lower N3 stage sleep are noteworthy and warrant further discussion in the discussion section.

·         The use of segmented regression models is a strength of this section, as it allows for a nuanced analysis of the relationship between EA and various sleep parameters.

3. Relationship Between EA and Sleep Outcomes (3.3):

·         The paper uses segmented regression models to detail the relationship between EA and sleep outcomes. Identifying breakpoints in this relationship is a valuable insight that could have significant implications for the field. However, it would be beneficial if the authors provided a more in-depth interpretation of these findings within the context of existing literature to highlight the novelty and significance of their results.

4. Interpretation and Context:

·         The results are presented straightforwardly; however, adding brief interpretations or implications of these findings after each subsection could enhance the reader's understanding and engagement.

5. Technical Corrections:

·         Ensure consistency in formatting, especially in tables and figures. Check for any typographical errors or inconsistencies in data presentation.

4. Discussion

·         While the discussion does a good job interpreting the findings, it could benefit from a more explicit statement on the broader implications for sports nutrition, coaching, and athlete health management.

·         Based on the study's findings, the discussion could offer more specific practical recommendations for athletic administrators, nutritionists, and coaches.

·         Further integrating the study's findings with the broader body of research on sleep, nutrition, and athletic performance would provide a more comprehensive understanding.

·         The discussion could be strengthened by addressing potential counterarguments or alternative interpretations of the data to demonstrate a thorough understanding of the study's context.

·         Ensure consistency and accuracy in citing literature and presenting data.

Comments on the Quality of English Language

The quality of English in the manuscript is good overall, with clear and well-structured sentences. The terminology is appropriate for the academic field, and the scientific concepts are conveyed effectively. There might be occasional minor issues or typographical errors that require attention, but these do not detract significantly from the overall readability or comprehension of the text.
